# Robust Metabolite Quantification from J-Compensated 2D ^1^H-^13^C-HSQC Experiments

**DOI:** 10.3390/metabo10110449

**Published:** 2020-11-07

**Authors:** Alexander Weitzel, Claudia Samol, Peter J. Oefner, Wolfram Gronwald

**Affiliations:** Institute of Functional Genomics, University of Regensburg, 93053 Regensburg, Germany; alexander.weitzel@physik.uni-regensburg.de (A.W.); claudia.samol@klinik.uni-regensburg.de (C.S.); peter.oefner@klinik.uni-regensburg.de (P.J.O.)

**Keywords:** metabolomics, NMR, quantification, HSQC, Q-HSQC, QUIPU-HSQC, water suppression, cryoprobe

## Abstract

The spectral resolution of 2D ${}^{1}$H-${}^{13}$C heteronuclear single quantum coherence (${}^{1}$H-${}^{13}$C-HSQC) nuclear magnetic resonance (NMR) spectra facilitates both metabolite identification and quantification in nuclear magnetic resonance-based metabolomics. However, quantification is complicated by variations in magnetization transfer, which among others originate mainly from scalar coupling differences. Methods that compensate for variation in scalar coupling include the generation of calibration factors for individual signals or the use of additional pulse sequence schemes such as quantitative HSQC (Q-HSQC) that suppress the *J*_CH_-dependence by modulating the polarization transfer delays of HSQC or, additionally, employ a pure-shift homodecoupling approach in the ${}^{1}$H dimension, such as Quantitative, Perfected and Pure Shifted HSQC (QUIPU-HSQC). To test the quantitative accuracy of these three methods, employing a 600 MHz NMR spectrometer equipped with a helium cooled cryoprobe, a Latin-square design that covered the physiological concentration ranges of 10 metabolites was used. The results show the suitability of all three methods for the quantification of highly abundant metabolites. However, the substantially increased residual water signal observed in QUIPU-HSQC spectra impeded the quantification of low abundant metabolites located near the residual water signal, thus limiting its utility in high-throughput metabolite fingerprinting studies.

## 1. Introduction

Measurement of the metabolism of an organism provides important insights into its functions. This is the goal of metabolomics, which aims at the comprehensive analysis of the entirety of metabolites in a biological system [1]. To this end, hyphenated mass spectrometry and nuclear magnetic resonance (NMR) spectroscopy are commonly used [2]. The suitability of NMR arises, among other factors, from the linear dependency of the signal volumes on the concentration values and the examination of a sample in a non-destructive fashion. For human metabolomics, the focus is often put on human body fluids such as blood, cerebral spinal fluid and urine [3]. Importantly, NMR spectra need to provide quantitative data beyond mere identification of compounds. The simplest way of quantifying compounds in a sample is the employment of 1D ${}^{1}$H NMR. However, this approach often shows limitations due to the presence of signal overlap that renders integration of signals difficult, if not impossible. There are several approaches for spectral deconvolution of 1D spectra such as Chenomx (Chenomx Inc. Edmonton, AB, Canada), Batman [4], Bayesil [5] and decon1d [6] available, which usually work well in the presence of moderate signal overlap. A suitable alternative is multidimensional NMR spectroscopy, such as 2D ${}^{1}$H-${}^{13}$C-HSQC, which reduces overlap by spreading the signals over two dimensions, but carries the disadvantage that the sole dependence of signal volume on substance concentration is lost [7,8]. Factors contributing to the loss of signal dependence on concentration are variations in the scalar coupling constant of the C–H bond (*J*_CH_) and the longitudinal and transversal relaxation times ($T_{1}$ and $T_{2}$) as well as the excitation profiles of the ${}^{13}$C-pulses. For quantification, the influence of these additional parameters needs to be corrected for. This can be realized by obtaining signal-specific calibration factors for serial dilutions of standards [8,9]. However, this is a time-consuming process, as for each quantifiable signal a separate calibration curve has to be built. Consequently, different pulse sequence schemes to account for the entirety or a portion of these factors have been developed. These include among others HSQC${}_{0}$ [10] and Q-HSQC [11]. In HSQC${}_{0}$, a series of HSQC spectra is acquired with different repetition times for the extrapolation of a time-zero HSQC in which signal intensities are directly proportional to concentrations. However, due to the acquisition of multiple HSQC spectra, the measurement times are significantly increased. In contrast, Q-HSQC accounts for the signal variations originating from variations in *J*_CH_ values by effectively averaging over different *J*_CH_ coupling constants, which is performed by acquiring 75% of the scans constituting a 2D increment with an insensitive nuclei enhancement by polarization transfer (INEPT) delay of $\Delta_{1}=2.94$ ms and the remaining 25% of the scans with an INEPT delay of $\Delta_{2}=5.92$ ms. This averaging leads to an approximately uniform magnetization transfer for *J*_CH_ values ranging from 110 to 220 Hz, which is shown by the red curve in Figure 1. Since its introduction, Q-HSQC has been steadily expanded and further developed [12,13,14,15,16]. A recent development is the QUIPU-HSQC experiment that provides, in addition, increased sensitivity by suppressing signal splitting caused by homo-nuclear proton-proton (H-H) coupling [15].

In this contribution, we systematically compare the quantification results obtained by standard 2D ${}^{1}$H-${}^{13}$C HSQC using individual calibration factors with both standard HSQC using J-compensation (Q-HSQC) and QUIPU-HSQC on a spectrometer equipped with a helium cooled cryoprobe. To this end, a Latin-square design comprising 10 different metabolites commonly found in human and spanning a concentration range from 10 to 5000 μM is employed. We address the following questions: (1) What is the accuracy and precision of the different methods and, most importantly, how do accuracy and precision change with decreasing metabolite concentration? (2) What is the general quality of the spectra in terms of the residual water signal and other artifacts? (3) Are these methods applicable to large scale metabolomic investigations in terms of measurement and data evaluation time?

## 2. Results

As detailed in the Materials and Methods section, a Latin-square design consisting of ten samples containing ten typical human metabolites, namely acetic acid, alanine, betaine, creatinine, citric acid, glycine, ethanolamine, histidine, taurine and trimethylamine-N-oxide, was employed.

### 2.1. Efficiency of Scalar Coupling Averaging

A central element of Q- and QUIPU-HSQC experiments is the reduction of the influence of one-bond ${}^{1}$H-${}^{13}$C scalar couplings on signal intensities. Therefore, we first simulated the relationship between these two factors. Figure 1 shows for a standard HSQC the expected relationship between the scalar coupling constants and signal intensity (blue curve). The distribution of coupling values for the ten investigated metabolites is indicated by red crosses. Clearly, for most signals, coupling values close or slightly below 150 Hz are expected and the corresponding variation in signal intensities is therefore also relatively small. However, for a few signals, considerably larger or smaller values were observed. In comparison, both the Q-HSQC and QUIPU-HSQC effectively remove for the range of observed coupling constants the dependence of signal intensities on coupling constants (orange line). However, this comes at the expense of reduced signal intensities for most of the signals.

The four histidine resonances C2 H2, C3 H3A/B, C5 H5 and C7 H7 are especially interesting signals for a detailed investigation of the efficiency of the scalar coupling compensation used by Q-HSQC and QUIPU-HSQC as their coupling constants range from 117 to 211 Hz (Table 1). For an easier comparison, all signal intensities were normalized to the C2 H2 signal and, therefore, are given as ratios. The column “Predicted Ratio” contains expected values for a standard HSQC optimized for a *J*_CH_ of 145 Hz. The next column shows the corresponding actually measured values from a standard HSQC without the application of correction factors, while the penultimate and ultimate columns show the observed ratios for the highest histidine concentration obtained by Q-HSQC and QUIPU-HSQC, respectively. As expected for standard HSQC spectra (Columns 3 and 4) a clear relationship between coupling values and signal intensities becomes apparent, while this is less the case for Q-HSQC. In this context, it is important to note that other factors that potentially affect signal intensities such as relaxation effects are still present in Q-HSQC. Interestingly, the QUIPU-HSQC obtained, especially for the C5 H5 and C7 H7 signals, considerably reduced values. This can be explained by the fact that for these two signals deteriorated signal-shapes were obtained that hindered proper signal integration. This is most likely the result of the applied homonuclear decoupling scheme of the QUIPU-HSQC. Central to the employed homonuclear decoupling is the BIRD element, which relies on the transfer of magnetization between carbons and protons and, therefore, as already pointed out by Zangger [17], is also affected by the large coupling values of the C5 H5 and C7 H7 resonances.

### 2.2. Accuracy and Precision

Next, the accuracy and precision of standard 2D ${}^{1}$H-${}^{13}$C HSQC using individual calibration factors were compared to those of Q-HSQC and QUIPU-HSQC, respectively. To this end, all samples of the latin-square design were measured in triplicate. To verify the concentration values of the Latin-square design the corresponding 1D ${}^{1}$H NMR spectra were analyzed as well. To correct for small pipetting errors during sample preparation, these values were set as true values for all following analyses. Note, as in this special case, where each sample contained 10 metabolites only, signal overlap normally hindering quantification by 1D NMR was not an issue. Figure 2 shows for the ten investigated metabolites the obtained quantification results as bar graphs. For the smallest five concentration levels results are also presented as magnified inserts. The relative difference between the concentration estimates and the true values, describing the accuracy, were below 15% in 71% of the concentration levels for the standard HSQC, in 69% for Q-HSQC, and in 35% for QUIPU-HSQC. As evident from the inserts in Figure 2, especially for QUIPU-HSQC larger deviations between true and measured values were obtained for the smaller concentrations. Next, for each 2D method, the relative error (RE) and relative standard deviation (RSD) were computed for each compound in each sample. Appendix A show the data for standard HSQC, Q-HSQC and QUIPU-HSQC, respectively. Taking alanine as a typical example, RE and RSD values of 4.0% and 1.0%, respectively, were obtained for the highest concentration level of 5.0 mM by the standard HSQC. At the lowest detectable concentration of 0.078 mM, RE and RSD increased to 19.0% and 7.0%, respectively. Over all investigated metabolites, average RE and RSD values of 3.9% and 0.9% were obtained for the highest concentration level, while, for a concentration of 0.078 mM, RE and RSD values increased to 21.1% and 6.7%, respectively.

For Q-HSQC, RE and RSD of 6.0% and 1.0%, respectively, were obtained for alanine at the 5.0 mM concentration level, while, for 0.078 mM, the RE and RSD were 5.0% and 19.0%, respectively. RE and RSD values averaged over all investigated metabolites amounted to 11.3% and 0.8% for the highest concentration level and 37.1% and 10.4% for 0.078 mM.

For QUIPU-HSQC, RE and RSD of 4.0% and 3.0%, respectively, were obtained for alanine at 5.0 mM, while, for 0.078 mM, no values could be determined due to insufficient detection sensitivity. For the next higher concentration level of 0.156 mM, the RE and RSD values amounted to 23.0 % and 8.0%, respectively. Averaged over all metabolites at the 5.0 mM concentration level, RE and RSD values of 13.4% and 4.1% were obtained. For a concentration level of 0.156 mM, the respective average values amounted to 140.7% and 28.6%, respectively.

Application of a Friedman test [18] to the results of the three investigated experimental approaches revealed for both REs and RSDs significant differences (*p* < 0.05). The Nemenyi post-hoc test (Appendix A) [19] revealed significant differences between the REs of standard HSQC and Q-HSQC (*p* < 0.05) as well as between Q-HSQC and QUIPU-HSQC (*p* < 0.05). Furthermore, the difference in REs between standard HSQC and QUIPU-HSQC is highly significant (*p* < 0.001). With respect to the RSDs, highly significant differences (*p* < 0.001) were obtained between QUIPU-HSQC and standard HSQC as well as QUIPU-HSQC and Q-HSQC. However, there was no significant difference between Q-HSQC and standard HSQC. To obtain a more detailed view on the differences between the three experiments, Friedman tests together with Nemenyi post-hoc tests were applied for each of the 10 investigated metabolites (Appendix A). Note that here we aim to investigate whether two methods perform equally well or not. Only in the absence of significant differences is the assumption of equal performance met. Therefore, to be as conservative as possible, no correction for multiple testing was performed. As can be seen from the third column of Appendix A, the Nemenyi post-hoc test revealed only for the REs of taurine and ethanolamine significant differences between the standard HSQC and the Q-HSQC. For RSDs, no significant differences between standard HSQC and Q-HSQC were observed. For the comparison of standard HSQC with QUIPU-HSQC significant differences in REs were obtained for acetic acid, TMAO and histidine, while the RSDs differed significantly (*p* < 0.05) for all metabolites except ethanolamine. A comparison of Appendix A shows that this may be explained by the generally increased RSDs of the QUIPU-HSQC. A comparison between Q-HSQC and QUIPU-HSQC in terms of REs reveals no significant differences, whereas significant RSD differences were observed for all metabolites except alanine and ethanolamine (Appendix A). As for the comparison of standard HSQC with QUIPU-HSQC, this may be explained by the generally increased RSDs of the QUIPU-HSQC.

### 2.3. Analysis of Matrix Effects

Given the superior performance of the Q-HSQC over the QUIPU-HSQC, the performance of the former was further investigated in a typical biological matrix. To this end, 21 amino acids were added at two different concentrations, namely at 300 and 600 μM each, to human urine obtained from a healthy volunteer. Next, Q-HSQC spectra were recorded for the blank and the two spike-in samples. To allow for an easier comparison between expected and obtained values, results are given in Appendix A as measured values minus blank values. For the 600 μM spike-in, 11 and 7 amino acids showed a deviation between observed and expect values of <10% and <20%, respectively, while for three amino acids deviations ≥20% were obtained. For the lower spike-in level of 300 μM, as expected, increased deviations of <10%, <20% and ≥20% were observed for six, seven and eight amino acids, respectively.

## 3. Discussion

We compared systemically the quantitative performance of three different 2D HSQC based methods, namely standard HSQC, Q-HSQC and QUIPU-HSQC, by employing a Latin-square design that comprised ten metabolites in a near to physiological concentration range of 10–5000 μM. All used pulse programs employed continuous wave water presaturation to achieve minimal disturbance of signals located in the vicinity of the residual water signal. Furthermore, adiabatic inversion pulses [20] were used on the ${}^{13}$C nuclei in order to minimize signal variations stemming from a limited excitation band width, as recommended by Koskela et al. [13]. Thus, the employed QUIPU-HSQC differed slightly from the published pulse sequence of Mauve et al. [15]. The results show that accurate and precise quantification of highly abundant metabolites might be achieved with all three 2D approaches. For concentrations of 500 μM and lower, which are found in human plasma and urine, the standard HSQC yielded at the 0.078 mM concentration level an average relative error of 21.1%, which increased to 37.1% for Q-HSQC. This increase in relative error in comparison to the standard HSQC is most likely due to the generally reduced sensitivity of the INEPT transfer of Q-HSQC, as shown in Figure 1.

For QUIPU-HSQC (Appendix A), only 7 out of 10 compounds could be detected at the 0.078 mM concentration level and for the next higher concentration level of 0.156 mM a very large average relative error of 140.7% was obtained. As for Q-HSQC, the INEPT transfer in QUIPU-HSQC is in most cases not as effective as in the standard HSQC (Figure 1). Furthermore, the employment of homonuclear bilinear rotation decoupling (BIRD) extends the experimental duration of QUIPU-HSQC and thus facilitates the recovery of the initially saturated water signal. Additionally, due to relaxation and scalar coupling evolution processes during the acquisition step, BIRD leads to sideband formation [21], which can only be partly compensated for by experimental means. In our case, where a helium cooled cryoprobe was employed, this led to a considerable increased and broadened residual water signal, which overlapped with several of the metabolite signals, thus impeding their accurate integration (Figure 3C). For example, the creatinine signal C2 H2A/B at 4.03/59.07 ppm overlaps with the residual water signal centered at around 4.78 ppm. It is known that water suppression in helium cooled cryoprobes can be difficult. Therefore, these observations are not unexpected. In the original QUIPU pulse sequence published by Mauve et al. [15], a presaturation scheme consisting of looped 180${}^{\circ }$ inversion selective shaped pulses flanked by cleaning gradients was employed. As this presaturation scheme gave in our case no satisfactory results it was replaced by continuous irradiation during the relaxation delay, which resulted in a slightly improved water suppression. Therefore, for all experiments presented in this study, water suppression was achieved by presaturation employing continuous irradiation.

Acetate represented an interesting exception. It was accurately quantified by standard 2D HSQC together with correction factors. However, both Q-HSQC and QUIPU-HSQC underestimated considerably the true amount of acetic acid present. The underestimation of acetic acid by these 2D ${}^{1}$H-${}^{13}$C-HSQC experiments originates from an insufficient recovery period. The longitudinal relaxation time of acetic acid protons is reportedly between 3.2 and 3.8 s [22,23] and thus significantly longer than that of other organic compounds of metabolic interest. The recovery period for all three 2D ${}^{1}$H-${}^{13}$C-HSQC was set to 6 s, which is sufficient for most compounds. This finding emphasizes the recommendation of Mauve et al. [15] to employ recovery periods of at least 15 s. However, as the total experimental time is mostly dictated by the recovery delay, an increase in recovery time from 6 to 15 s would increase the total measurement time by a factor of 2.5, which is infeasible in high-throughput routine metabolic applications. Therefore, as a compromise, we stayed with a recovery delay of 6 s. Note that the calibration factors employed together with the standard HSQC also corrected for differences in longitudinal relaxation.

We also investigated the performance of the Q-HSQC in a real biological matrix by adding a set of 21 amino acids at two different levels to a matrix of human urine. The results show that for a spike-in level of 600 μM reliable results could be obtained with only two amino acids exceeding deviations between expected and measured values by more than 20%. However, this number increased to eight for a reduced spike-in level of 300 μM. In the Latin-square design, where 10 metabolites were added to a matrix of pure water, this corresponds to spike-in level 6 (312 μM and Appendix A). Here, only for acetate a relative error larger than 20% was obtained (Appendix A). This difference is on one hand explained by the decreased sensitivity of a cryoprobe in the presence of salt as contained in a typical biofluid such as human urine and on the other hand by the additional error stemming from the concentration estimation in the blank urine sample.

## 4. Materials and Methods

### 4.1. Samples

A Latin-square design consisting of ten samples was prepared to evaluate the experimental performance of the standard HSQC, Q-HSQC and QUIPU-HSQC. The ten samples consisted of geometrically diluted solutions of acetic acid, alanine, betaine, citric acid, creatinine, ethanolamine, glycine, histidine, taurine and trimethylamine-N-oxide, which were chosen under considerations of spectral overlap, biological importance and variations in *J*_CH_ coupling. The ten dilution levels ranged from 5000 to 10 μM, whereby the overall substance concentration in each sample was equal. Further information about the composition of the samples can be found in Appendix A.

For the three urine samples, 200 μL of human urine were mixed with either 200 μL of pure water (blank sample) or with 200 μL of an aqueous mixture of 21 amino acids so that for the two spike-in samples the added concentrations for each amino acid amounted to 300 and 600 μM, respectively. The used urine was obtained from one patient in the context of the German Chronic Kidney Disease (GCKD) Study. The study was conducted in accordance with the Declaration of Helsinki and it was approved by the local ethics committees and registered in the national registry for clinical studies (DRKS 00003971). All study procedures and protocols were approved by the ethics committees of all participating institutions (Friedrich-Alexander-University Erlangen-Nuremberg, Medical Faculty of the Rheinisch-Westfälische Technische Hochschule Aachen, Charité-University Medicine Berlin, Medical Center-University of Freiburg, Medizinische Hochschule Hannover, Medical Faculty of the University of Heidelberg, Friedrich-Schiller-University Jena, Medical Faculty of the Ludwig-Maximilians-University Munich, Medical Faculty of the University of Würzburg). The study was carried out in accordance with relevant guidelines and regulations. Written declarations of informed consent had been obtained from all study participants before inclusion.

### 4.2. Sample Preparation

NMR samples with a total volume of 650 μL were prepared, consisting of 200 μL of 0.1 M phosphate buffer with a pH value of 7.4, which contained in addition 3.9 mM boric acid to impair the growth of bacteria in the sample; 50 μL of 0.75 (wt) TSP in deuterium oxide (D${}_{2}$O), which serves as the internal quantification standard; and 400 μL metabolite mix. For the 10 Latin-square samples, the metabolite mix corresponds in each case to one of the 10 different mixes of the Latin-square design. For the urine samples, 400 μL of the three different urine mixes were used.

### 4.3. NMR Spectroscopy

1D ${}^{1}$H NOESY, 2D ${}^{1}$H-${}^{13}$C-HSQC, 2D Q-HSQC and 2D QUIPU-HSQC NMR experiments were performed at 298 K on a Bruker Avance III spectrometer (Bruker BioSpin GmbH, Rheinstetten, Germany) utilizing a triple resonance (${}^{1}$H, ${}^{13}$C, ${}^{15}$N, ${}^{2}$H lock) cryogenic probe with *z*-gradients in combination with a Bruker SampleJet sample changer (Bruker BioSpin GmbH, Rheinstetten, Germany). Prior to measurement, thermal equilibration of the sample was allowed for 300 s before automatically locking, tuning, matching and shimming the probe. A description of the used NMR parameters along with the details of the pulse programs used are available in the supporting information.

### 4.4. Data Evaluation

All 2D spectra were semi-automatically processed with TopSpin3.1 (Bruker BioSpin GmbH, Rheinstetten, Germany) employing a manual phase and a polynomial baseline correction, excluding the region around the water artifact. All spectra were chemical shift referenced relative to the TSP signal. 1D spectra were quantified using Chenomx 8.5 (Chenomx Inc. Edmonton, AB, Canada). Signal integration of all 2D spectra was performed using AMIX 3.9.13 (Bruker BioSpin GmbH, Rheinstetten, Germany). For quantification from standard HSQC spectra, in-house obtained calibration factors [8] together with the quantification tool MetaboQuant [24] were used. For Q-HSQC and QUIPU-HSQC, which did not require individual calibration factors, concentrations were directly obtained with respect to the integral of the TSP reference signal. Note that the independence from variations in proton–carbon couplings comes at the price of generally reduced signal intensities (Figure 1). However, as this affects both metabolite signals and the reference signal similarly, this effect cancels out. A summary of the signal properties of the ten investigated metabolites is provided in Appendix A. For statistical data evaluation, Friedman and Nemenyi post-hoc tests were performed in *R* v.4.0.0 (The R Foundation for Statistical Computing, Vienna, Austria).

## 5. Conclusions

We showed that, for the metabolites present at high abundance >1 mM, the guidelines of the Food and Drug Administration [25]requiring for bioanalytical methods a RSD of <15% regarding precision and a RE of <15% regarding accuracy are mostly met by all three investigated 2D methods (see Appendix A), hence accurate and precise quantification is facilitated by all three of them. For lower abundant metabolites, both standard HSQC and Q-HSQC are viable choices. The clear advantage of the Q-HSQC is despite a minor loss of sensitivity the feasibility of accurate quantification without the need of additional calibration factors. This facilitates its routine application. Table 2 summarizes the advantages and limitations of the three different methods.

As shown in the table, the standard HSQC takes 55 min in measurement time, while this amount doubles for both the Q-HSQC and the QUIPU-HSQC, which is due to the doubled relaxation delay of 6 s required for the latter two experiments. For large-scale metabolomics investigations, this results in 24 samples that can be measured with the standard HSQC on a single day, while this amount halves for both Q-HSQC and QUIPU-HSQC. However, with the latter two experiments, a considerable amount of time is saved as no signal-specific calibration factors have to be determined.

Although not tested here, QUIPU-HSQC can be expected to perform best as long as water suppression is not required. However, this will entail more elaborate sample preparation such as extraction of metabolites and their reconstitution in deuterium oxide. Recent approaches of fast 2D NMR spectra acquisition by non-uniform sampling [26], use of variable recycling times [27], spectral aliasing [28] and its direct combination with Q-HSQC and QUIPU-HSQC [16] can further facilitate quantitative 2D NMR spectra acquisition in a single experiment. However, the same limitations as shown here will apply to the quantification of analytes.

## Figures and Tables

**Figure 1 metabolites-10-00449-f001:**
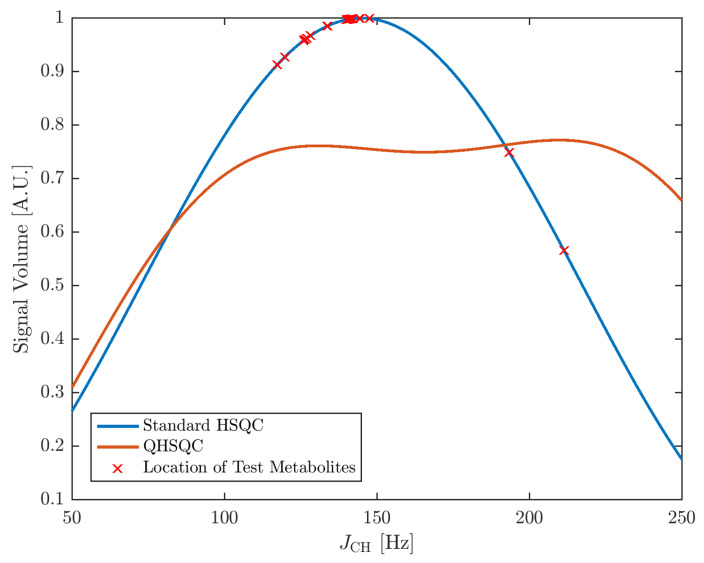
Dependency of ${}^{1}$H-${}^{13}$C-HSQC cross-peak signal volumes on the scalar coupling constant *J*_CH_ for a standard HSQC experiment with an INEPT delay Δ=3.40 ms in blue and a Q-HSQC experiment with INEPT delays of $\Delta_{1}=2.94$ and $\Delta_{2}=5.92$ ms, respectively, at a ratio of 3:1 in orange. The crosses indicate the *J*_CH_-coupling values of 20 signals of the metabolites in the used model mixture (see Materials and Methods section for details).

**Figure 2 metabolites-10-00449-f002:**
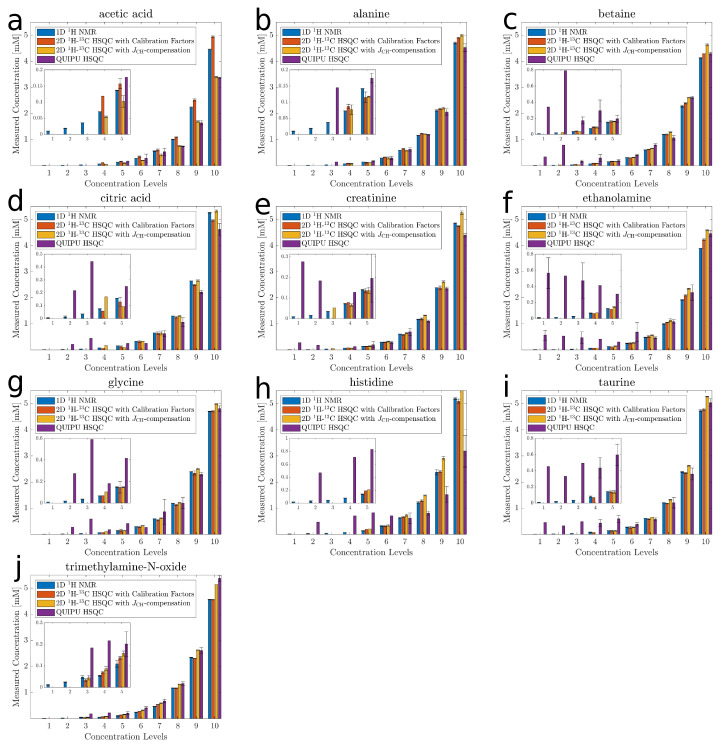
Comparison of concentration estimates for acetic acid (**a**), alanine (**b**), betaine (**c**), citric acid (**d**), creatinine (**e**), ethanolamine (**f**), glycine (**g**), histidine (**h**), taurine (**i**), and trimethylamine-N-oxide (**j**). The *x*-axis shows the ten concentration levels and the *y*-axis the observed concentrations with 1D ${}^{1}$H NMR in blue, standard HSQC in red, Q-HSQC in orange and QUIPU-HSQC in purple. The lower five concentration levels are added as inserts. The concentration values are means over technical triplicates. In the case that the compound was detected in all three replicates, the corresponding standard deviation is included as a measure of error.

**Figure 3 metabolites-10-00449-f003:**
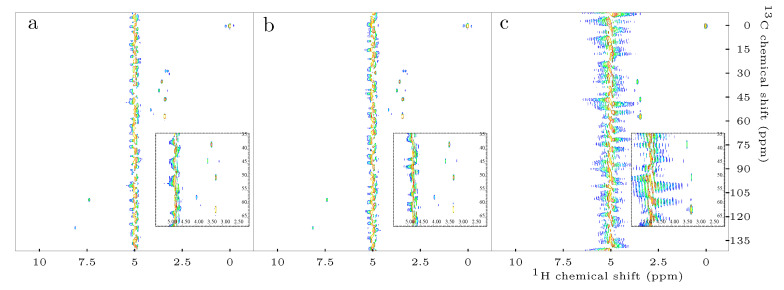
Exemplary 2D ${}^{1}$H-${}^{13}$C-HSQC spectra obtained by means of: standard HSQC (**a**); Q-HSQC (**b**); and QUIPU-HSQC (**c**). Inserts depict the spectral regions of 70.0–35.0 and 5.5–2.0 ppm. A considerable number of signals stemming from aliphatic protons of the investigated metabolites is located in this region. Furthermore, the residual water signal at 4.78 ppm is located in this region, thus frequent overlap is present.

**Table 1 metabolites-10-00449-t001:** Integral volumes of the four histidine signals normalized to C2H2. The third column shows the predicted ratios modulated by non-uniform magnetization transfer due to different *J*_CH_ values. The predicted ratios were calculated on the basis of in-house measured *J*_CH_ values. The fourth column presents the measured ratios for the highest histidine concentration measured by standard HSQC without correction by calibration factors. The penultimate and ultimate columns show the observed ratios for the highest histidine concentration observed by Q-HSQC and QUIPU-HSQC, respectively.

	Coupling Constant [Hz]	Predicted Ratio	Without Averaging	Q-HSQC	QUIPU-HSQC
C2 H2	140.8	1	1.00 ± 0.4%	1.00 ± 1%	1.00 ± 4%
C3 H3A/B	128.3/117.4	0.94	0.97 ± 2%	0.87 ± 2%	0.83 ± 2%
C5 H5	193.4	0.75	0.82 ± 2%	1.01 ± 0.8%	0.54 ± 19%
C7 H7	211.4	0.57	0.61 ± 5%	0.93 ± 3.6%	0.48 ± 6%

**Table 2 metabolites-10-00449-t002:** Summary of the properties of Standard HSQC, Q-HSQC and QUIPU-HSQC. LODs and LLOQs were calculated from the results in Appendix A. The LOD is defined as the minimum concentration above which a compound is always detected. The LLOQ is defined according to FDA guideline as the minimum concentration for which the RSD is <20%.

	Standard HSQC	Q-HSQC	QUIPU-HSQC
mean LOD [μM]	78	80	71
mean LLOQ [μM]	187	148	640
Requirement of calibration values	Yes	No	No
Sensitivity towards ${}^{1}$H-${}^{13}$C coupling	Yes	No	No
Suppression of ${}^{1}$H-${}^{1}$H couplings	No	No	Yes
Sensitivity towards residual water signal	Normal	Normal	Increased
Measurement time [min]	55	108	110

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
