# Peer review of "Robust Metabolite Quantification from J-Compensated 2D ^1^H-^13^C-HSQC Experiments"

_metabolites, 2020, doi:10.3390/metabo10110449_

Round 1

Reviewer 1 Report

dear authors, i really enjoyed reading your manuscript. 

I only have two comments

  1. According to the ICH Q10/FDA  guide for bio-analysis at the LLOQ You can accept a precision and/or accuracy  20 % RSD, 15 %over LLOQ. You mostly achieve this.
  2. 2. You specify that You prepare Your samples using 0,1 mM  (line 342, page 8 of 10) phosphate buffer conditioned with 0.3 mM boric aid buffer. I doubt that Your buffer capacity is sufficient to keep chemical shifts stable especially for urine samples. Our experience that it takes up to 0.3 M to stabilize chemical shifts in urine.  You might also want to condition with EDTA or fluoride to reduce variation from metal ions such as mg2+ or Ca 2+.  Azide ions or  Fluoride ions would be another possibility to impair bacterial growth.

Otherwise, good work

Reviewer 2 Report

The manuscript by Weizel et al. presents an investigation of three different two-dimensional (2D) NMR experiments employed on a urine sample spiked with 10 different metabolites in different concentration ranges. The aim of the study is to examine the accuracy and precision of the three different methods, and also to elucidate how applicable they are to large scale metabolomic investigations. The manuscript is well written and clear, and the methodologies applied appears sound and solid. It is also evaluated that the topic addressed is of interest for the scientific audience working in the area of NMR-based metabolomics. Overall, the study can be considered to represent a relevant contribution to the scientific community. I only have minor comments that the authors may consider:

Introduction, lines 26-28: It is correct that spectral overlapping can occur in 1D and that this is a challenge. Nevertheless, several applications and softwares providing tools for spectral deconvolution are available, and it would be relevant to mention this here to make the text more nuanced.

Line 216: ‘…were present at high abundance’ : It is suggested to specify the concentration range that the methodology appears to be viable.

Lines 215-226: Possibly the advantages and limitations for the three different methods mentioned here could be summarized in a Table?

Lines 215-226: The text does not really answer question no 3 raised in the introduction: Are these methods applicable to large scale metabolomic investigations in terms of measurement and data evaluation time? (lines 57-58)

Reviewer 3 Report

In this study, the authors explore in deep the use of the HSQC method in the identification and quantification of metabolites based on NMR spectroscopy. The article also provides an extensive comparison between the variants of the HSQC method, namely Q-HSQC and QUIPU-HSQC.  This work is very enlightening on metabolomics analysis of biofluids and comes from a research group with long-lasting experience on the subject.

A reasonable question that could be asked is how the quantitative data of the metabolites obtained by the QUIPU-HSQC method, were adequately compared with the values obtained from the 1D-HNMR spectrum, since the QUIPU-HSQC method is based on the reduction of constant coupling, but which results in reduced signal intensities of the metabolites. And also, it would be valuable to use a larger number of metabolites, as well as metabolites of lower molecular weight than those used, to see the effectiveness of the Q-HSQC method, since it is claimed to be the most effective of the HSQC variants.
